# State Design Matters: How Representations Shape Dynamic Reasoning in Large Language Models

**Annie Wong**                                                                *a.s.w.wong@liacs.leidenuniv.nl*
*Leiden Institute of Advanced Computer Science, Leiden University*

**Aske Plaat**
*Leiden Institute of Advanced Computer Science, Leiden University*

**Thomas Bäck**
*Leiden Institute of Advanced Computer Science, Leiden University*

**Niki van Stein**
*Leiden Institute of Advanced Computer Science, Leiden University*

**Anna V. Kononova**
*Leiden Institute of Advanced Computer Science, Leiden University*

**Reviewed on OpenReview:** *https://openreview.net/forum?id=sKoazMNH84&noteId=zcqXCre2jj*

## Abstract

As large language models (LLMs) move from static reasoning tasks toward dynamic environments, their success depends on the ability to navigate and respond to an environment that changes as they interact at inference time. An underexplored factor in these settings is the representation of the state. Holding model parameters fixed, we systematically vary three key aspects: (1) state granularity (long form versus summary), (2) structure (natural language versus symbolic), and (3) spatial grounding (text-only versus images or textual map encodings) across sequential decision-making benchmarks. We find that *trajectory summarisation* improves performance by reducing noise and stabilising long-horizon reasoning. Second, *natural language* representations are the most robust across models, whereas structured encodings help mainly for models with strong code or structured output priors, such as JSON schemas. Third, while image-inputs show some benefit, *text-based spatial encodings* prove most effective. This advantage stems not from the spatial information itself, but from the act of construction, which compels the model to perform the spatial reasoning that static input does not elicit. Overall, we demonstrate that design choices for representing state are a decisive factor in performance, distinct from the availability of information itself. We note, however, that even with improved representations, current LLMs and VLMs remain brittle over long horizons, particularly when they must synthesise information to manage multiple subtasks to reach a goal.

Our code is available at `https://tinyurl.com/state-representations`.

## 1 Introduction

Benchmarks for dynamic, multi-step reasoning have rapidly developed in recent years, reflecting a broader shift toward evaluating large language models (LLMs) in interactive settings where they act as controllers (Paglieri et al., 2025; Wu et al., 2024b). Yet evaluation practice remains dominated by static, single-turn benchmarks, such as question answering, coding, and math word problems (Brown et al., 2020; Chowdhery et al., 2023; Wei et al., 2022b). Strong performance on these tasks has fueled claims that reasoning abilities emerge with scale (Wei et al., 2022a). However, whether such behaviours constitute genuine reasoning

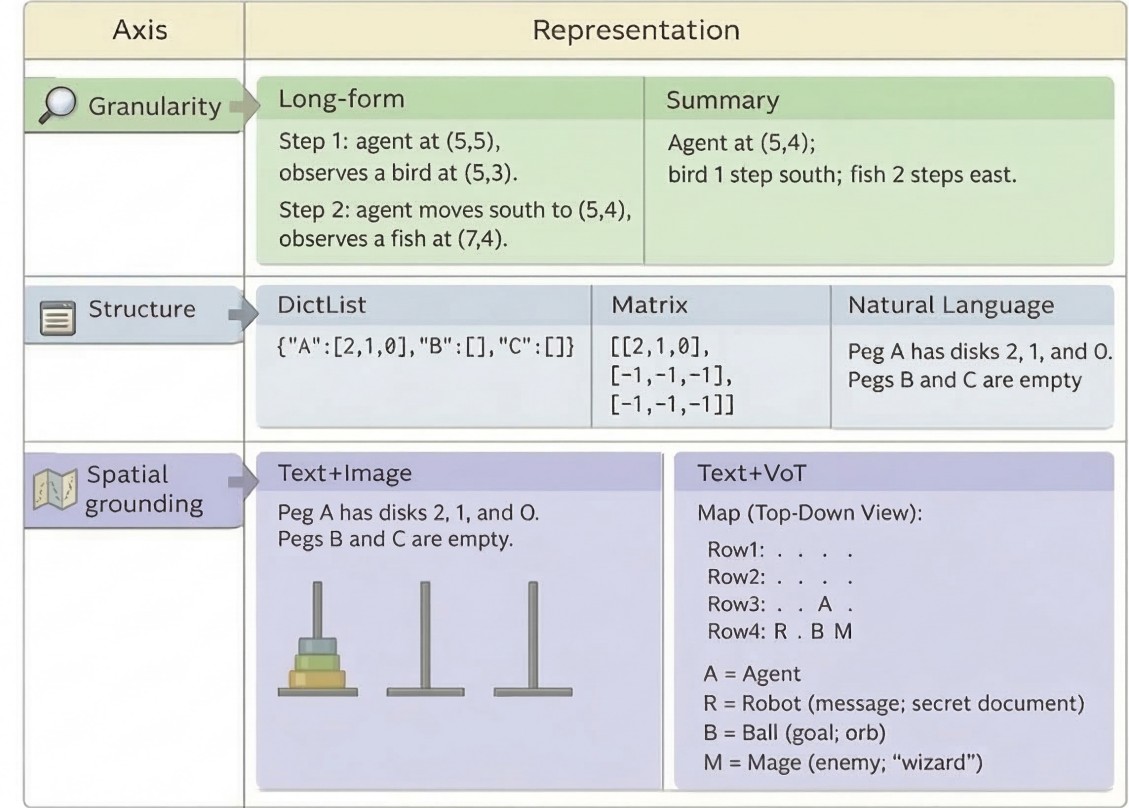

Figure 1: State representation variants evaluated in our study. The design space is organised along three axes: granularity (long-form trajectories vs. summaries), structure (natural language vs. concise symbolic encodings), and spatial grounding (text-only vs. explicit grounding via images or structured text). These axes capture key prompt-level design choices that influence how the environment state is presented to the model, enabling a systematic study of how trajectory compression, structural constraints, and spatial grounding affect long-horizon decision-making.

remains contested (Mitchell, 2025). Crucially, static benchmarks decouple reasoning from interaction. They do not test whether a model can revise its beliefs and decisions as the environment changes in response to its actions, which is a core requirement of autonomous agents (Wong et al., 2025).

In sequential decision-making, it is well established that agent performance depends on the quality of the state representation, and that poorly designed encodings can impair decision making and control (Lesort et al., 2018). Dynamic LLM benchmarks instantiate the same representational problem: the agent must repeatedly condition its actions on an evolving state expressed through text. Yet, despite the growing use of LLMs as controllers, these representational trade-offs have received little systematic study.

At inference time, LLM-based agents receive the environment state through an externally constructed prompt. Although descriptive natural language aligns with common pretraining data, it introduces three distinct challenges as interactions lengthen. We discuss (1) granularity, (2) structure and (3) modality of the prompt representation.

The first challenge is verbosity. Natural language descriptions rapidly expand the token count of the prompt, causing performance to suffer when critical information is lost in the middle of long contexts, a phenomenon where models fail to uniformly attend to all parts of a lengthy input (Liu et al., 2024). To mitigate this, previous work has employed context compression via summarisation (Li et al., 2023). However, for summarisation to be effective, task-critical information must remain preserved; in dynamic, long-horizon settings, omitting or distorting relevant details can alter downstream decisions and compound errors over future steps.

The second challenge is ambiguity and lack of structure. Natural language is expressive but often imprecise for state tracking: values may be implicit, ordering may vary, and the same state can be paraphrased in many ways. This forces the model to infer how the environment changed. One alternative is to impose structure through symbolic, schema-constrained encodings (e.g., dictionaries, lists, or matrices) that make state variables explicit and easier to update consistently after each action. For instance, consider the Tower of Hanoi state described in text as *"Peg A has disks 2, 1, and 0 on it, from bottom to top. Peg B and Peg C are both empty"*; this can be encoded as a dictionary, *"A": [2,1,0], "B": [], "C": []*. Such representations can improve planning in some settings (Hu et al., 2024), though evidence for consistent benefits across tasks and model families remains mixed (Sullivan & Elsayed, 2025).

The third challenge is spatial grounding. When environments require spatial reasoning, natural language may obscure geometric relations under verbose descriptions. While multimodal LLMs can consume visual inputs (Masry et al., 2022; Lu et al., 2023; Yue et al., 2024), spatial reasoning remains underexplored (Shiri et al., 2024) and gains are inconsistent (Paglieri et al., 2025; Yue et al., 2024). "Visualization-of-Thought" (VoT), a prompting method that elicits spatial reasoning by having models visualize intermediate reasoning traces, suggests that pixel inputs are not always necessary: structured text layouts such as ASCII grids can be sufficient for spatial tasks (Wu et al., 2024a).

Together, these challenges motivate a systematic investigation of how prompt-level state design influences long-horizon control: whether compressing trajectories, imposing structure, or adding spatial grounding improves performance, and under what task and model conditions these choices help or hurt. Therefore, we ask the following research question:

*How does inference-time state representation, along the axes of granularity (long-form trajectories versus summaries), structure (natural language versus symbolic encodings), and spatial grounding (text-only representations versus explicit spatial grounding via images or textual map encodings), affect performance on dynamic reasoning tasks?*

We evaluate these design choices in multi-step, interactive environments where the state evolves in response to the agent's actions and success requires long-horizon planning, memory, and spatial reasoning. We consider a range of open-source LLMs spanning small to large scales, focusing on in-context learning with frozen parameters and no gradient updates or fine-tuning.

Our main contributions are as follows:

- Using a *summary* of the interaction history, rather than the full history, can improve long-horizon decision-making by removing noise, helping the agent to focus on task-relevant information. Performance effects are task- and model-dependent. Summarisation tends to help when the task-relevant state can be captured by a compact abstraction, the summary is accurate, and contains key information for action selection.

- Across tasks, *natural language* is the most robust state representation; when structured encodings help, gains are largely confined to models with stronger exposure to code or structured outputs, which can reliably exploit JSON-like schemas, whereas other models fail to correctly interpret the structured encodings.

- Adding a spatial representation to the prompt can improve performance, and among variants *Visualization of Thought* is the most consistently effective (Wu et al., 2024a). However, these gains depend on sufficient model capability to construct faithful maps. Moreover, we find that VoT primarily helps by inducing step-by-step spatial reasoning rather than by supplying a richer state description.

The rest of our paper is organised as follows. Section 2 provides the background, Section 3 details the methodology, Section 4 presents the results, and Section 5 concludes with a discussion of the findings.

# 2 Background

## 2.1 State representation

In sequential decision-making, a state representation specifies how the environment is encoded for a policy, with the goal of retaining task-relevant information while discarding irrelevant variation. In particular, a good state representation should satisfy four criteria (Lesort et al., 2018; Böhmer et al., 2015):

1. The state is Markovian: the state summarizes all information necessary for the agent to select an action based on the current state alone.

2. The state representation contains sufficient information to assess the quality (value) of the current state and judge which actions are likely to lead to a successful outcome.

3. The representation supports generalisation to unseen states with similar future outcomes.

4. The representation remains low dimensional to enable efficient decision-making.

Whether a given state encoding actually meets these goals cannot be inferred from its format; it must be tested by evaluating the performance of a controller that uses it (Lesort et al., 2018). Since multiple encodings can work for the same task, there is no single best representation; the right choice depends on the task and the controller.

We next summarise state representations used in prior work, organised by textual encodings and spatial grounding.

## 2.2 State granularity

Prior work in reinforcement learning suggests that compressing observations into language-based state representations can improve generalisation and interpretability, and that representation design can matter even when the model is held fixed (Rahman & Xue, 2024; Echchahed & Castro, 2025).

Closely related to our work, recent research in dynamic routing games studies natural language state representations along three axes: action informativeness, reward informativeness, and the degree of natural language compression (Goodyear et al., 2025). The study demonstrates that restricting information granularity improves stability. Specifically, they find that agents reach game-theoretic equilibrium more effectively when provided with history summaries, regret feedback, and limited visibility of others' actions. However, the study is limited to natural language, a single task, and one language model. In contrast, we study how granularity, structure, and modality interact across multiple tasks and different model architectures.

## 2.3 Structure and symbolic representations

A common way to improve reasoning is by eliciting intermediate reasoning steps (Plaat et al., 2026), as in Chain-of-Thought (Wei et al., 2022b), Self-Refine (Madaan et al., 2023), and Reflexion (Shinn et al., 2023). While this can improve performance, it often becomes verbose over long horizons and may add redundancy or ambiguity. A complementary line of work uses structured or symbolic encodings instead. Methods such as Chain-of-Symbol represent state in symbolic forms, reducing token usage and improving planning in some settings (Hu et al., 2024). Related approaches use symbolic structure for verification or tool use (Chen et al., 2024b; Pan et al., 2023), or switch between symbolic and natural language formats depending on the subproblem (Han et al., 2024).

Despite these advances, it remains unclear how to trade off structured encodings versus natural language use in dynamic environments where the state evolves continuously and the agent must maintain a consistent representation across a long history of interactions.

### 2.4 Spatial grounding

Beyond purely textual representations, recent work explores whether grounding state information in perceptual form can improve spatial reasoning. Vision–language models (VLMs), which accept images alongside text, enable agents to condition decisions on visual input, providing a direct mechanism for spatial grounding.

On static vision benchmarks, adding real images can be beneficial. For example, on MathVista, GPT-4V substantially outperforms its text-only counterpart, indicating that access to raw visual input can improve mathematical reasoning in visual contexts (Lu et al., 2023). However, results are more mixed in dynamic settings: in sequential environments that require repeated state updates and long-horizon planning, conditioning on raw images can be ineffective or even harmful, with VLM agents underperforming text-only variants (Paglieri et al., 2025).

Several methods introduce spatial grounding without relying on raw images. Visualization-of-Thought prompts models to generate structured textual diagrams, such as ASCII grids, that explicitly encode spatial layouts (Wu et al., 2024a), while Visual Chain-of-Thought extracts salient visual regions and integrates them into a textual reasoning trace (Shao et al., 2024). Rather than adding sensory input, these approaches externalise the environment's spatial structure in a form that remains directly accessible to the language model throughout inference.

Most methods are evaluated in static, single-turn settings, where the model receives a complete instance (e.g., a full map and instructions) and produces a solution in one pass. While this probes multi-step reasoning, it sidesteps the central challenge of interactive control, where the agent must update its state representation after every action as the environment changes. We build on this line of work by comparing image-conditioned and text-conditioned grounding in dynamic environments to understand when spatial grounding facilitates effective sequential decision-making.

## 3 Methodology

### 3.1 Markov decision process

We model each environment as a Markov decision process over discrete timesteps $t \in \{1, \ldots, T\}$. An episode starts in an initial state $\boldsymbol{s}_1$ and ends when a terminal condition is reached. At each timestep $t$, the agent receives an observation $o_t$; in fully observed settings $o_t = \boldsymbol{s}_t$, while in partially observed settings $o_t$ provides only partial information about $\boldsymbol{s}_t$. The agent then selects an action $\boldsymbol{a}_t$ according to a stochastic policy $\pi(\boldsymbol{a}_t \mid x_t)$, where in our setting $\pi$ is instantiated by a fixed, pre-trained LLM conditioned on the inference-time prompt $x_t$. After executing $\boldsymbol{a}_t$, the environment returns a scalar reward $r_t$ and transitions to the next state $\boldsymbol{s}_{t+1}$ according to the dynamics $P(\boldsymbol{s}_{t+1} \mid \boldsymbol{s}_t, \boldsymbol{a}_t)$. The goal within an episode is to maximise the cumulative reward $R = \sum_{t=1}^{T} r_t$, where $T$ denotes the episode horizon.

### 3.2 State representation

We use the inference-time prompt as the state interface. Because the model maintains no persistent internal state across timesteps, all information required for decision-making must be provided explicitly at each step. At timestep $t$, we construct a prompt

$$x_t = [\mathcal{M}; \phi(o_t, \hat{h}_{t-1}); \mathcal{A}],$$

where $\mathcal{M}$ is a static task manual, $\phi(o_t, \hat{h}_{t-1})$ encodes the current observation together with a history representation, and $\mathcal{A}$ lists the available actions. We keep $\mathcal{M}$ and $\mathcal{A}$ fixed within an environment and vary only the encoding produced by $\phi$.

The current observation $o_t$ describes the instantaneous environment configuration under a chosen encoding format. Depending on the environment, $o_t$ may be fully observed or partial.

We organise the prompt-level state representation design space along three axes: granularity (full history vs. summary), structure (free-form natural language vs. symbolic encodings), and spatial grounding (text-only vs. explicit grounding via images or structured text layouts).

**Granularity** controls how the history is provided to the LLM via $\hat{h}_{t-1}$. We compare two settings:

$$\textsc{Long Form:} \quad \hat{h}_{t-1} = h_{t-1},$$

$$\textsc{Summary:} \quad \hat{h}_{t-1} = m_{t-1}, \qquad m_t = \psi(m_{t-1}, o_t, a_t, r_t),$$

where $h_{t-1} = (o_1, a_1, r_1, \ldots, o_{t-1}, a_{t-1}, r_{t-1})$ is the full interaction history and $m_t$ is a rolling natural-language summary updated each step by a fixed summarisation procedure $\psi$ implemented with the same LLM, where the summarisation instruction explicitly requests a summary of at most 25 tokens. The summary replaces the full history in the prompt to reduce context length while preserving task-relevant information.

For instance, when we have the full past trajectory in the prompt, it might look like below:

```
Past trajectory:
Step 1: You took action stay. You (agent) don't have the message.
        You see:
        - bird 2 steps away
        - sword 1 step away
        - fish 3 steps away
Step 2: You took action east. You (agent) already have the message.
        You see:
        - bird 2 steps away
        - fish 4 steps away
```

When summary is enabled, this history is compressed into a summary that the LLM updates at each timestep, replacing the full trajectory in the prompt. For the same sequence above, the summary might read:

```
Summary of past actions:
Moved east, collected message from sword; bird 2 steps away, fish 4 steps away.
```

We note that the Summary condition introduces an additional LLM call per timestep to produce $m_t$. While the input context to the agent is shorter (since $m_{t-1}$ replaces $h_{t-1}$), the total tokens consumed by the system increase once summarisation tokens are included, since the SUMMARY condition adds an extra LLM call that takes the full history $h_{t-1}$ (and the previous summary $m_{t-1}$) as input to produce the compressed state $m_t$.

**Structure** specifies the state is encoded in the prompt. On the one hand, we consider free-form natural language descriptions. On the other hand, we consider structured encodings, including key–value dictionaries, matrices, grid-based layouts, and hybrid formats. Structured representations constrain how information is organised, which can reduce ambiguity and make parsing easier, but may limit expressive flexibility. Natural language representations retain semantic detail and contextual nuance, but leave structure implicit and require the model to infer it. For instance, Hanoi can be represented via natural language like:

```
Peg A has disks 2, 1, and 0 (bottom to top). Pegs B and C are empty.
```

or in matrix form like:

```
[[2,1,0],
[-1,-1,-1],
[-1,-1,-1]]
```

**Spatial grounding** determines whether the state is provided purely as text or augmented with explicit spatial grounding signals. In addition to text-only representations, we consider inputs that include rendered images of the environment as well as textual visualisations that encode spatial layouts directly, via Visualization-of-Thought, where ASCII sketches are generated as part of the agent's reasoning process (Wu et al., 2024a). These variants differ in whether spatial structure is conveyed through an image or through structured text, and may interact differently with model pretraining and capacity.

For instance, the state in Hanoi might be represented purely via text:

Table 1: Environment specifications for all experiments. We report task difficulty, history length (number of past steps maintained in context), rollout length (episodes per run), action-space size, and maximum timesteps per episode. Runs are reported as S/M(L), where S and M denote the number of runs for small and medium models, respectively, and the value in parentheses denotes the number of runs for large models. In our setup, LLAMA3.3-70B and QWEN3-VL-32B-INSTRUCT are classified as large models; the remaining models are classified as small or mid-sized.

| Environment | Difficulty | History | Rollout | Actions | Timesteps | Runs |
|---|---|---|---|---|---|---|
| Hanoi | Medium | 30 | 10 | 6 | 30 | 10 (5) |
| Messenger | Hard | 10 | 20 | 5 | 10 | 10 (5) |
| BabyAI-GoTo | Easy | 128 | 10 | 6 | 128 | 25 (10) |
| BabyAI-Open | Medium | 128 | 10 | 6 | 128 | 25 (10) |
| BabyAI-PickUp | Medium | 128 | 10 | 6 | 128 | 25 (10) |
| BabyAI-PickUpSeqGoTo | Hard | 128 | 10 | 6 | 128 | 25 (10) |
| BabyAI-PutNext | Hard | 128 | 10 | 6 | 128 | 25 (10) |

```
Peg A has disks 2, 1, and 0. Pegs B and
C are empty.
```

or via a textual "visualisation of thought" that makes the spatial layout explicit, where the numbers refer to the size of the disks.

```
Map (Top-Down View):
Rod A: [2, 1, 0]  (top is 0, bottom is 2)
Rod B: []
Rod C: []
```

Spatial grounding can additionally be provided through rendered images of the agent's partial observation. In this vision-language setting, the prompt includes both the textual state description and an image depicting the current view, allowing models to recover spatial relations directly from images rather than from symbolic text alone.

Figure 1 provides illustrative examples for each axis. Full prompts and examples are provided in the Appendix.

### 3.3 Test environments

We evaluate three dynamic environments: Tower of Hanoi and Messenger from SmartPlay (Wu et al., 2024b), and BabyAI from BALROG (Paglieri et al., 2025). All environments require multi-step planning and spatial reasoning. Tower of Hanoi requires moving three disks across three pegs while respecting the constraint that no larger disk may be placed on a smaller one. Messenger is a grid-world task where the agent must pick up a message and reach a goal while avoiding an enemy. Entities are described using synonyms to test semantic robustness. BabyAI consists of five navigation and interaction tasks of increasing difficulty (Chevalier-Boisvert et al., 2019; Carta et al., 2023). The agent operates in a 2D grid-world and follows natural-language missions, ranging from simple navigation (`GoTo`) and door interaction (`Open`) to object manipulation (`PickUp`, `PutNext`) and short compositional instructions (`PickUpSeqGoTo`).

BALROG provides both text and image observations, whereas SmartPlay is text-only. To support our spatial grounding experiments, we implemented an image renderer for SmartPlay that produces a visualisation of each state.

### 3.4 Experimental setup

We evaluate a set of open-source LLMs to study how model scale and architecture affect performance. The models were selected to cover a range of parameter scales and architectural families, including both

text-only and vision–language models. This allows us to examine whether the observed representation effects are consistent across heterogeneous models rather than being specific to a single architecture. For text-only representations, we use PHI4-14B, LLAMA3.1-8B, DEEPSEEK-R1-14B, and LLAMA3.3-70B. For multimodal representations (vision + text), we use QWEN2.5VL-7B, LLAVA-PHI3-3.8B, LLAVA-7B, and QWEN3-VL-32B-INSTRUCT. We run the largest models, LLAMA3.3-70B and QWEN3-VL-32B-INSTRUCT, on two machines equipped with NVIDIA A100 GPUs; the remaining models run on NVIDIA RTX 4090, RTX 3060, or A40 GPUs. All experiments use temperature 0.2 with nucleus sampling (top_p = 0.95) to reduce stochastic variability while avoiding fully deterministic decoding.

### 3.5 Scoring

Performance is evaluated using the task-specific episode score $s_e$ at the end of evaluation episode $e$. In Tower of Hanoi, $s_e$ is the number of disks placed on the goal peg, taking values in $[0, 3]$. In Messenger, $s_e$ is the episode reward, which lies in $[-1, 1]$. In BabyAI, $s_e$ is binary, with 1 indicating successful task completion and 0 otherwise.

For each run, we average scores over $E$ evaluation episodes,

$$s_{\mathrm{run}} = \frac{1}{E} \sum_{e=1}^{E} s_e,$$

where $E$ is the number of evaluation episodes per run. We then report the mean over $R$ runs,

$$\bar{s} = \frac{1}{R} \sum_{r=1}^{R} s_{\mathrm{run},r},$$

where $r$ indexes runs and $s_{\mathrm{run},r}$ denotes the mean score of run $r$.

To enable comparison across environments, we linearly rescale the mean score $\bar{s}$ to a normalized score $z \in [0, 1]$, where 0 denotes the worst outcome and 1 denotes successful task completion. For Tower of Hanoi, we use $z = \bar{s}/3$, and for *Messenger*, $z = (\bar{s} + 1)/2$. BabyAI scores already lie in $[0, 1]$, so no additional normalization is required. Standard deviations are transformed using the same linear scaling.

For all comparisons against the baseline, we assess significance using a run-level bootstrap test of the mean difference. For each run, we first average the benchmark-specific episode scores to obtain a single run-level score. We then generate 10,000 bootstrap replicates by resampling these per-run scores with replacement and compute percentile 95% confidence intervals for the mean difference, and significance markers are based on Holm-corrected two-sided bootstrap p-values.

## 4 Results

We present results for each of the three state-representation axes introduced in Section 3.2: granularity (full trajectory vs. summary), structure (natural language vs. symbolic encodings), and spatial grounding (text-only vs. images or textual map encodings). For each axis, we report performance, highlight recurring patterns and failure modes, and conclude with an overall summary of the main findings.

### 4.1 Granularity

We study how the granularity of the historical trajectory in the agent prompt affects performance. For these experiments, we use SmartPlay's default state representations: *TaggedList* for Tower of Hanoi and *NaturalLanguage* for Messenger (see Appendix for the format). In *Long Form*, the prompt includes the full interaction history $h_{t-1}$, whereas in SUMMARY the history is replaced by a rolling summary $m_{t-1}$ updated at each time step.

Table 2: Performance under *Long Form* versus *Summary* prompting on Tower of Hanoi and Messenger across open-source LLMs. Entries report mean normalised score $\pm$ SD over runs ($n$=10 seeds per model–environment setting, except LLAMA3.3-70B with $n$=5), with scores scaled to $[0, 1]$ where 1 denotes task completion. A superscript $*$ marks a significant difference from the baseline (Long Form) for the same model and task, using a run-level bootstrap test of the mean difference over per-run scores (10,000 resamples; percentile 95% CI). Two-sided bootstrap p-values are adjusted with the Holm procedure across all pairwise comparisons reported within this table, controlling the family-wise error rate at $\alpha = 0.05$.

Overall, summarisation tends to help Hanoi for mid-to-large models (notably QWEN2.5VL-7B, LLAMA3.1-8B, DEEPSEEK-R1-14B, and QWEN3-VL-32B-INSTRUCT), but is less reliable in Messenger, where it produces large gains for some models (e.g., DEEPSEEK-R1-14B and LLAMA3.1-8B) while degrading others (e.g., QWEN2.5VL-7B and QWEN3-VL-32B-INSTRUCT).

| | Hanoi | | Messenger | |
|---|---|---|---|---|
| Model | Long Form | Summary | Long Form | Summary |
| LLaVA-Phi3-3.8B | 0.16 (0.12) | 0.08 (0.10) | 0.24 (0.11) | 0.14 (0.06)$^*$ |
| LLaVA-7B | 0.00 (0.10) | 0.14 (0.16) | 0.26 (0.10) | 0.31 (0.05) |
| Qwen2.5VL-7B | 0.08 (0.09) | 0.39 (0.00)$^*$ | 0.11 (0.06) | 0.05 (0.13) |
| Llama3.1-8B | 0.03 (0.09) | 0.25 (0.10)$^*$ | 0.20 (0.13) | 0.35 (0.07)$^*$ |
| Phi4-14B | 0.37 (0.17) | 0.44 (0.11) | 0.34 (0.25) | 0.44 (0.36) |
| DeepSeek-R1-14B | 0.46 (0.12) | 0.70 (0.21)$^*$ | 0.53 (0.28) | 1.00 (0.17)$^*$ |
| Qwen3-VL-32B-Instruct | 0.81 (0.07) | 0.98 (0.03)$^*$ | 0.09 (0.09) | 0.00 (0.13) |
| Llama3.3-70B | 0.66 (0.47) | 1.00 (0.00) | 0.28 (0.19) | 0.22 (0.05) |

Table 3: Performance under *Long Form* vs. SUMMARY prompting on five BabyAI tasks (multimodal models only). Entries report mean normalised score $\pm$ SD over runs ($n$=25 seeds per model–task setting, except QWEN3-VL-32B-INSTRUCT with $n$=10), with scores scaled to $[0, 1]$ where 1 denotes task completion. A superscript $*$ marks a significant difference from the baseline (Long Form) for the same model and task, using a run-level bootstrap test of the mean difference over per-run scores (10,000 resamples; percentile 95% CI). Two-sided bootstrap p-values are adjusted with the Holm procedure across all pairwise comparisons reported within this table, controlling the family-wise error rate at $\alpha = 0.05$.

Overall, summarisation most reliably helps in simpler navigation tasks (e.g., GoTo) and can improve performance in compositional tasks (e.g., PickUp and PickUpSeqGoTo), but offers limited benefit in harder tasks (Open and PutNext).

| | GoTo | | PickUp | | PickUpSeqGoTo | | Open | | PutNext | |
|---|---|---|---|---|---|---|---|---|---|---|
| Model | Long Form | Summary | Long Form | Summary | Long Form | Summary | Long Form | Summary | Long Form | Summary |
| LLaVA-Phi3-3.8B | 0.28 (0.46) | 0.52 (0.51) | 0.04 (0.20) | 0.12 (0.33) | 0.08 (0.28) | 0.08 (0.28) | 0.00 (0.00) | 0.00 (0.00) | 0.00 (0.00) | 0.00 (0.00) |
| LLaVA-7B | 0.72 (0.46) | 0.44 (0.51) | 0.12 (0.33) | 0.20 (0.41) | 0.12 (0.33) | 0.16 (0.37) | 0.00 (0.00) | 0.00 (0.00) | 0.00 (0.00) | 0.04 (0.20) |
| Qwen2.5VL-7B | 0.88 (0.33) | 0.96 (0.20) | 0.24 (0.44) | 0.24 (0.44) | 0.16 (0.37) | 0.32 (0.48) | 0.00 (0.00) | 0.00 (0.00) | 0.00 (0.00) | 0.00 (0.00) |
| Qwen3-VL-32B-Instruct | 0.60 (0.52) | 0.60 (0.52) | 0.00 (0.00) | 0.10 (0.32) | 0.10 (0.32) | 0.30 (0.48) | 0.80 (0.42) | 0.40 (0.52) | 0.00 (0.00) | 0.00 (0.00) |

### 4.1.1 Hanoi

Summarisation is beneficial for most models in Tower of Hanoi (Table 2). With the exception of the smallest model (LLAVA-PHI3-3.8B, which degrades under compression from 0.16 to 0.08), all larger models improve under *Summary*, with gains increasing with model scale. For example, QWEN2.5VL-7B improves from 0.08 to 0.39*, DEEPSEEK-R1-14B from 0.46 to 0.70*, and QWEN3-VL-32B-INSTRUCT from 0.81 to 0.98*. The largest model, LLAMA3.3-70B, reaches perfect task completion under *Summary*, improving from 0.66 to 1.00. This pattern is consistent with Hanoi admitting a compact abstraction: once the current disk configuration is preserved, earlier trajectory details are largely redundant, so summarisation primarily removes noise while retaining the task-critical state.

### 4.1.2 Messenger

Table 2 reports the Messenger results. Unlike Hanoi, the effect of summarisation is mixed: some models benefit substantially, while others are unchanged or degrade. In particular, DEEPSEEK-R1-14B achieves a perfect score, improving from 0.53 to 1.00*, LLAMA3.1-8B from 0.20 to 0.35*, and PHI4-14B from 0.34 to 0.44. In contrast, several models regress under SUMMARY, including QWEN3-VL-32B-INSTRUCT (from 0.09 to 0.00) and LLAMA3.3-70B (from 0.28 to 0.22).

The summarisation failures largely depend on what information is retained. Models often produce generic summaries. For example, LLAMA3.1-8B writes *"Agent has not obtained the message yet, and there are entities nearby (scientist, ship, robot)"*, while omitting critical details such as relative distances, outcomes of recent actions, or progress toward subgoals. Failures also arise from reasoning limitations, such as choosing to "do nothing" even when the message holder is one step away. Qualitative traces further indicate that the benefit of summarisation is model-specific and depends on whether the model can maintain a coherent internal state. For instance, with DEEPSEEK-R1-14B on Messenger, adding summaries stabilises the trajectory: without summaries the agent oscillates (*"Move North $\to$ Move West $\to$ Move East $\to$ Move West"*) and finishes with score 0, whereas with summaries (e.g., *"Agent moves north toward the dog to get the message"*) the same model reaches score 1 in a short rollout.

Overall, summarisation shows more mixed results in Messenger. While a summary can help by focusing on the important parts, it causes the agent to fail if it leaves out any of the critical details, and it cannot compensate for the model's reasoning limitations.

### 4.1.3 BabyAI

Table 3 shows the BabyAI results for multimodal models. Summarisation is most beneficial in navigation-dominated settings, while gains diminish as tasks require more precise instruction following and relational reasoning. In GoTo, weaker models can improve under *Summary* (e.g., LLaVA-Phi3-3.8B: 0.28 to 0.52), but this effect is not uniform: LLaVA-7B collapses under *Summary* (0.72 to 0.44). By contrast, larger models achieve good performance under *Long Form* and receive little benefit from *Summary* (e.g., Qwen2.5VL-7B: 0.88 to 0.96; Qwen3-VL-32B-Instruct: 0.60 to 0.60).

In PickUp, where the agent must navigate to a specified object and pick it up, summarisation provides small gains for weaker models (e.g., LLaVA-Phi3-3.8B: 0.04 to 0.12; LLaVA-7B: 0.12 to 0.20), but does not help stronger models (Qwen2.5VL-7B: 0.24 to 0.24). This is consistent with summaries helping mainly by retaining the target identity and approximate location during exploration, rather than improving the interaction itself.

In the PickUpSeqGoTo task, summarisation yields modest but consistent improvements (e.g., LLaVA-7B: 0.12 to 0.16; Qwen2.5VL-7B: 0.16 to 0.32), while LLaVA-Phi3-3.8B remains unchanged (0.08 to 0.08).

In contrast, in harder settings, summarisation offers limited benefit and can even harm performance. In Open, most models remain at 0 regardless of representation, and Qwen3-VL-32B-Instruct degrades under *Summary* (0.80 to 0.40), suggesting that compression can be harmful when success depends on precise instruction execution. Similarly, PutNext remains near zero across all models and both representations, indicating that summarisation does not resolve the underlying demands of the task. Overall, summarisation helps most when the task state can be captured as simple spatial progress, but provides limited value as instruction complexity and relational constraints increase.

Qualitative inspection suggests that task success depends on the summary correctness. High-quality summaries act as efficient memory by preserving goal locations and progress, whereas weaker models sometimes hallucinate state updates (e.g., reporting that an object was picked up when it was not), which can trigger premature termination because the agent incorrectly believes (sub)goals have already been completed.

Table 4: Performance under *Long Form*, Summary and Oracle Summary on Tower of Hanoi. Entries report mean normalised score $\pm$ SD over 5 runs, with scores scaled to $[0, 1]$ where 1 denotes task completion. Oracle Summary provides the summary extracted programmatically rather than produced by an LLM. A superscript $*$ marks a significant difference from *Long Form* for the same model and environment, using a bootstrap test of the mean difference over per-run scores (10,000 resamples; 95% CI); to address multiple comparisons, p-values are adjusted with the Holm–Bonferroni method ($\alpha = 0.05$) across all pairwise comparisons, controlling the family-wise error rate.

Overall, Llama3.1-8B and DeepSeek-R1-14B perform best under Oracle Summary, indicating performance loss from imperfect LLM-generated summaries, whereas LLaVA-Phi3 collapses under Oracle Summary, suggesting limited ability to act on a compressed state interface even when the summary is correct.

| Environment | LLaVA-Phi3 | | | LlaMA3.1-8B | | | DeepSeek-R1-14B | | |
|---|---|---|---|---|---|---|---|---|---|
| | Long Form | Summary | Oracle Summary | Long Form | Summary | Oracle Summary | Long Form | Summary | Oracle Summary |
| Hanoi (3-disk) | 0.20 (0.08) | 0.16 (0.02) | 0.00 (0.00)$^*$ | 0.15 (0.03) | 0.23 (0.04)$^*$ | 0.27 (0.01)$^*$ | 0.35 (0.08) | 0.49 (0.10)$^*$ | 0.57 (0.08)$^*$ |

### 4.1.4 Ablation: summary oracle

The *Summary* condition requires the model to summarise its own interaction history, which conflates two factors: whether a compressed state is an effective interface for control, and whether the model can reliably produce such a summary.

To disentangle these effects, we introduce an *Oracle Summary* ablation on Tower of Hanoi, where the environment admits an exact Markov state description: a summary that specifies the current configuration is sufficient for action selection as the optimal next move depends only on the present peg–disk arrangement, not on the trajectory history. In the standard *Summary* condition, the rolling summariser converts this structured state into a natural-language description (e.g., "Peg 1 has disks 3, 2, and 1 from bottom to top; pegs 2 and 3 are empty."). In *Oracle Summary*, we bypass LLM summarisation and provide the acting agent with the ground-truth current configuration and goal directly from the simulator. This experiment isolates whether performance gains from rolling summaries arise from presenting state information in a more usable form, or whether LLM-generated summaries introduce noise that degrades control.

Table 4 shows that the benefit of compression is model-dependent. For Llama3.1-8B, performance improves from 0.15 *Long Form* to 0.23 *Summary* and further to 0.27 *Oracle Summary*. DeepSeek-R1-14B shows the same pattern, improving from 0.35 to 0.49 and to 0.57. These gains indicate that both models can exploit a perfectly compressed state, and that the gap between *Summary* and *Oracle Summary* is attributable to information loss or distortions introduced during summary generation.

In contrast, LLaVA-Phi3 does not benefit from summarisation: it drops from 0.20 *Long Form* to 0.16 *Summary* and collapses under *Oracle Summary* (0.00). This suggests a different limitation: even when the summary is accurate, some models struggle to interpret and act on it.

### 4.1.5 Cross-task comparison

Across tasks, Fig. 2 shows that *Summary* typically improves *score per input token* by keeping the acting prompt short, which reduces long-horizon prompt noise and focuses the agent on task-relevant state.

Performance effects, however, are task- and model-dependent. Summarisation tends to help when the task-relevant state can be captured by a compact abstraction, the summary is faithful and accurate, and contains all information needed for action selection, which is more likely in simple navigation tasks than in settings that require tracking relations and completing multi-step subgoals. Finally, in the most challenging environments, differences between *Long Form* and *Summary* become difficult to interpret: poor performance reflects a task-level capability bottleneck rather than a representational trade-off.

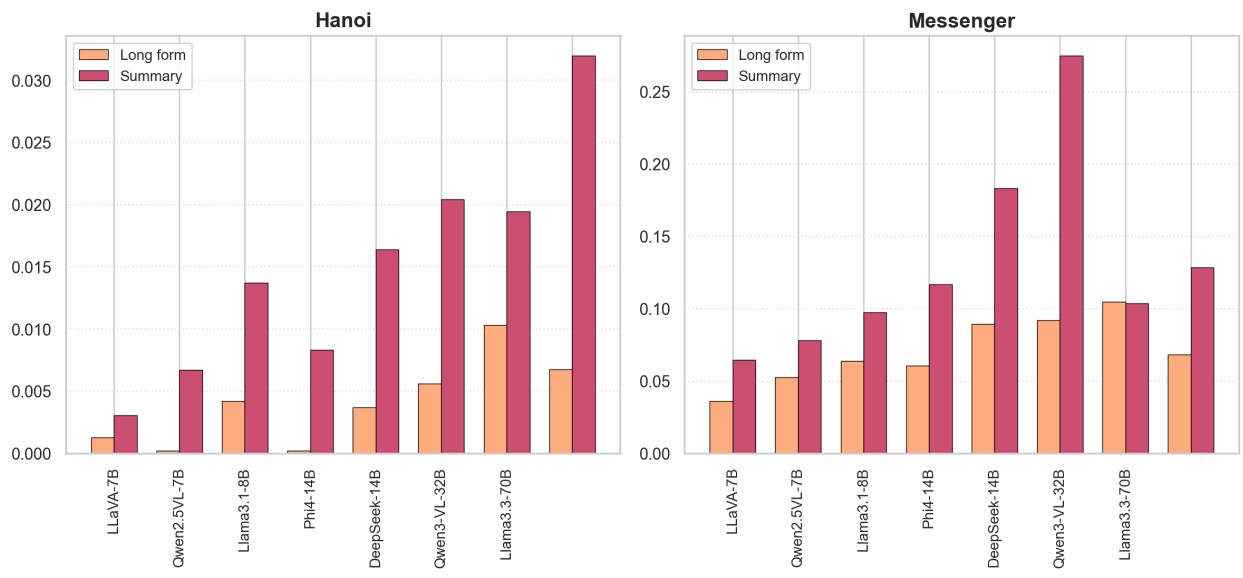

(a) Tower of Hanoi and Messenger.

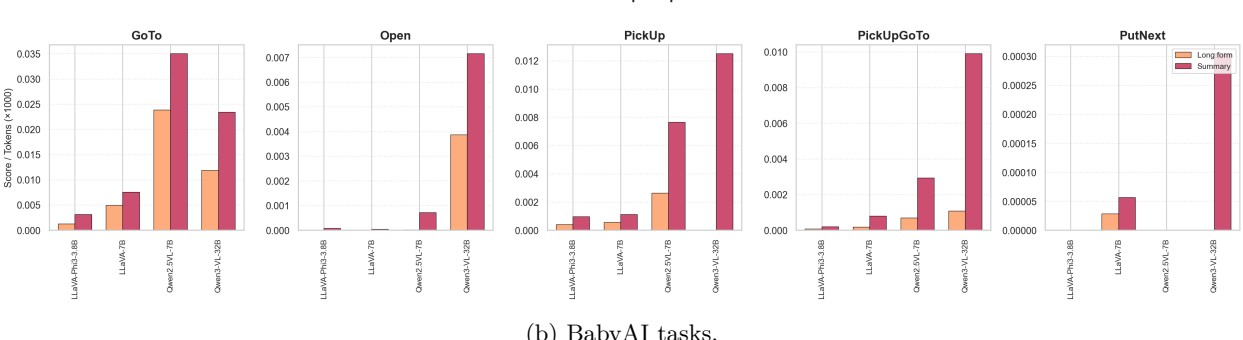

(b) BabyAI tasks.

Figure 2: Normalized score per input token for *Long Form* vs. *Summary* prompting. Bars report normalised score per input token ($\times 1000$), computed as the episode score (normalised to 0–1) divided by the average input prompt tokens. Higher values indicate better performance per token spent. The *Long Form* condition provides the acting agent with the full interaction history at each step, whereas *Summary* replaces this history with a rolling summary intended to preserve task-relevant state while reducing context length. Empty bars indicate settings where the agent achieves zero score. Token counts include only the input prompt shown to the acting agent. The *Summary* condition additionally requires a separate summarisation call, those tokens are excluded to isolate the effect of input context length on decision quality (i.e., we are not measuring end-to-end efficiency here.)

## 4.2   Structure

We next study how the *structure* of the prompt-level state representation affects agent performance. We treat *NaturalLanguage* as the reference format and compare it to symbolic encodings, ranging from semi-structured encoding (e.g., tagged lists or natural-language augmented with coordinates) to fully symbolic formats (e.g., dictionaries or symbolic grid layout). All results are reported in Table 5; examples of each format are provided in the Appendix.

Table 5: Performance under alternative state structures on Tower of Hanoi and Messenger across open-source LLMs. Entries report mean normalised score $\pm$ SD over runs ($n$=10 seeds per model–representation setting, except LLAMA3.3-70B with $n$=5), with scores scaled to $[0,1]$ where 1 denotes task completion. A superscript $*$ marks a significant difference from the *NaturalLanguage* baseline for the same model and environment, using a bootstrap test of the mean difference over per-run scores (10,000 resamples; 95% CI); to address multiple comparisons, p-values are adjusted with the Holm–Bonferroni method ($\alpha = 0.05$) across all pairwise comparisons, controlling the family-wise error rate. In Hanoi, dictionary-style structure (*DictList*) can help models that reliably interpret the schema, while grid-like *Matrix* encodings are often brittle and frequently degrade performance. In Messenger, structure offers limited upside and symbolic encodings tend to hurt, suggesting that preserving linguistic cues is critical for maintaining entity state and resolving synonym variation. A notable exception is DEEPSEEK-R1-14B, which benefits from explicitly providing the agent's own position (*NaturalLanguagePos*) that uses this information to guide its actions, whereas other models cannot.

**Hanoi**

| Model | NaturalLanguage | TaggedList | Matrix | DictList |
|---|---|---|---|---|
| Structure | Natural Language | Hybrid | Symbolic | Symbolic |
| LLaVA-Phi3-3.8B | 0.25 (0.05) | 0.21 (0.06) | 0.24 (0.08) | 0.31 (0.09) |
| LLaVA-7B | 0.12 (0.04) | 0.12 (0.05) | 0.00 (0.00)$^*$ | 0.01 (0.02)$^*$ |
| Qwen2.5VL-7B | 0.33 (0.00) | 0.16 (0.05)$^*$ | 0.33 (0.00) | 0.33 (0.00) |
| Llama3.1-8B | 0.13 (0.07) | 0.13 (0.05) | 0.03 (0.03)$^*$ | 0.08 (0.07) |
| Phi4-14B | 0.33 (0.10) | 0.32 (0.09) | 0.36 (0.09) | 0.33 (0.08) |
| DeepSeek-R1-14B | 0.33 (0.12) | 0.37 (0.07) | 0.31 (0.10) | 0.45 (0.18) |
| Llama3.3-70B | 0.67 (0.00) | 0.48 (0.26) | 0.22 (0.08)$^*$ | 0.27 (0.04)$^*$ |

**Messenger**

| Model | NaturalLanguage | NaturalLanguagePos | Coordinates | Symbolic |
|---|---|---|---|---|
| Structure | Natural Language | Hybrid | Hybrid | Symbolic |
| LLaVA-Phi3-3.8B | 0.43 (0.06) | 0.23 (0.06)$^*$ | 0.42 (0.04) | 0.36 (0.05) |
| LLaVA-7B | 0.44 (0.05) | 0.40 (0.04) | 0.42 (0.06) | 0.29 (0.08)$^*$ |
| Qwen2.5VL-7B | 0.36 (0.03) | 0.39 (0.08) | 0.38 (0.05) | 0.31 (0.10) |
| Llama3.1-8B | 0.41 (0.07) | 0.39 (0.06) | 0.42 (0.05) | 0.41 (0.06) |
| Phi4-14B | 0.48 (0.13) | 0.39 (0.03) | 0.43 (0.09) | 0.40 (0.09) |
| DeepSeek-R1-14B | 0.58 (0.15) | 0.75 (0.10)$^*$ | 0.44 (0.09) | 0.39 (0.09)$^*$ |
| Llama3.3-70B | 0.45 (0.10) | 0.37 (0.07) | 0.39 (0.05) | 0.40 (0.05) |

### 4.2.1 Hanoi

In Tower of Hanoi, state *structure* can matter, but its value is highly model-dependent. Relative to *NaturalLanguage*, symbolic encodings make peg–disk assignments explicit and can reduce relational inference, benefiting models that reliably map syntax to task constraints.

The semi-structured *TaggedList* format yields mixed effects: performance drops for QWEN2.5VL-7B (0.33 to 0.16$^*$), leaves LLAMA3.1-8B unchanged (0.13 to 0.13), and produces only small changes for LLAVA-7B and DEEPSEEK-R1-14B. This suggests that hybrid list syntax with natural-language cues does not consistently align with the patterns models exploit.

Dictionary encodings, equivalent to a JSON representation, can help when the model reliably grounds symbols to task constraints. Under *DictList*, LLAVA-PHI3-3.8B improves (0.25 to 0.31) and DEEPSEEK-R1-14B improves from (0.33 to 0.45). Qualitative analysis indicates that successful agents exploit explicit structural cues—such as empty lists ([])—to rule out illegal moves before planning. However, other models drop in performance: LLAVA-7B collapses (0.12 to 0.01$^*$) by repeatedly selecting moves from empty pegs, and LLAMA3.3-70B degrades markedly (0.67 to 0.27$^*$), suggesting failures to interpret the JSON-like schema.

*Matrix* encodings are the least reliable and often harmful (LLaVA-7B: 0.12 to 0.00*; Llama3.1-8B: 0.13 to 0.03*; Llama3.3-70B: 0.67 to 0.22*), consistent with an added decoding burden and recurring misreads (e.g., treating $-1$ as a valid disk instead of a padding value). For example, in Hanoi under *Matrix*, Llama3.1-8B repeatedly proposes moves from the empty rod B (*"Action: 4 ... move top of rod B to rod C"*).

Overall, *NaturalLanguage* remains the most robust choice across models, and the best-performing large model (Llama3.3-70B) achieves its top Hanoi score under *NaturalLanguage* (0.67). Structured formats can yield gains for specific model families (notably *DictList* for DeepSeek-R1-14B), but benefits are not universal, implying that representation design should be matched to model-specific strengths.

### 4.2.2 Messenger

In Messenger, the impact of representational structure is weaker than in Hanoi, and symbolic encodings are largely ineffective. Unlike Hanoi, which is strictly rule-governed, Messenger requires maintaining entity references, synonyms, and relationships over time. As a result, representations that abstract away linguistic cues can discard the semantic information models rely on to infer roles and intent.

*NaturalLanguagePos* adds the agent's own coordinates to the natural language prompt. The effect is model-dependent: DeepSeek-R1-14B improves (0.58 to 0.75*), while LLaVA-Phi3-3.8B drops (0.43 to 0.23*); most others change modestly, suggesting that added spatial detail can reduce ambiguity for higher-capacity models but distract smaller ones.

The fully *Symbolic* encoding typically degrades performance (LLaVA-Phi3-3.8B: 0.43 to 0.36; LLaVA-7B: 0.44 to 0.29*; DeepSeek-R1-14B: 0.58 to 0.39*), consistent with qualitative failures to map abstract state variables back onto the manual-defined entity roles. Agents fail to recognize which object corresponds to the enemy, the message, and the goal, leading to incorrect role assignments and downstream action choices.

*Coordinates* is generally closer to the *NaturalLanguage* baseline (e.g., LLaVA-Phi3-3.8B: 0.43 to 0.42; Llama3.1-8B: 0.41 to 0.42) but can still hurt (DeepSeek-R1-14B: 0.58 to 0.44). Qualitative analysis reveals that deriving directionality from raw coordinates remains difficult for LLMs. We find that agents often fail to recognize the correct trajectory and instead select actions that increase the distance to the goal.

In the Messenger environment, models that rely on explicit reasoning traces often perform best when observations are expressed in *NaturalLanguage*. For example, given the description *"You see a bird two steps west, a fish two steps north, and a sword two steps south"*, DeepSeek-R1-14B correctly identifies the sword as the message and reasons that moving south will reach it. The textual description directly exposes entity names and relative spatial relations, allowing the model to reason over the environment in a way that resembles natural language instructions.

### 4.2.3 Cross-task comparison

Across both tasks, *NaturalLanguage* remains the most robust representation. It is consistently competitive across model families. That said, when structure helps, it is most reliably provided by *DictList*. In Hanoi, *DictList* can reduce relational inference by making constraints explicit, which some models exploit to avoid illegal moves. These interactions appear correlated with a model's ability to treat the prompt as a structured data interface. Models with stronger evidence of code or structured-output training (e.g., DeepSeek-R1-14B (Guo et al., 2025) and Qwen2.5VL-7B (Yang et al., 2024)) are generally more robust to JSON-like schemas, while others degrade under the same format, suggesting schema interpretation failures rather than task difficulty. Technical reports for the Qwen2.5 family emphasize coding, structured data extraction, JSON-style outputs, and tool use (Yang et al., 2024), which may make Qwen-based models better able to interpret prompts such as *DictList*. Consistent with this view, DeepSeek-R1-14B, distilled from Qwen2.5, may inherit similar structured-output priors. By contrast, documentation for Llama 3.1 (Grattafiori et al., 2024), and LLaVA (Liu et al., 2023) emphasizes dialogue, reasoning, or multimodal instruction following rather than explicit schema-oriented outputs.

Table 6: Effect of spatial grounding variants on Tower of Hanoi, Messenger, and BabyAI across multimodal models. Entries report mean normalised score $\pm$ SD over runs, with $n$=10 seeds for Hanoi and Messenger (except QWEN3-VL-32B-INSTRUCT with $n$=5) and $n$=25 runs per BabyAI configuration (except QWEN3-VL-32B-INSTRUCT with $n$=10), and scores scaled to $[0, 1]$ where 1 denotes task completion. A superscript * marks a significant difference from the *Summary* (text-only) baseline for the same model and environment, using a bootstrap test of the mean difference over per-run scores (10,000 resamples; 95% CI); to address multiple comparisons, p-values are adjusted with the Holm–Bonferroni method ($\alpha = 0.05$) across all pairwise comparisons, controlling the family-wise error rate.

We compare this baseline to augmentations that add either an image (*Vision*) or a structured text-based spatial map (*VoT*). Shading is normalised within each game family. *Vision* is generally inconsistent and often degrades performance, suggesting that images rarely provide useful additional grounding beyond the text state. In contrast, *VoT* more frequently improve results, especially in navigation-focused BabyAI tasks, though gains are not universal and depend on the model's ability to construct and use the map reliably.

| Environment | LLaVA-Phi3-3.8B | Qwen2.5VL-7B | LLaVA-7B | Qwen3-VL-32B-Instruct |
|---|---|---|---|---|
| Hanoi Summary | 0.16 (0.05) | 0.33 (0.00) | 0.19 (0.09) | 0.65 (0.02) |
| Hanoi Summary + Vision | 0.11 (0.05) | 0.33 (0.00) | 0.25 (0.10) | 0.47 (0.22) |
| Hanoi Summary + VoT | 0.18 (0.04) | 0.38 (0.05)* | 0.39 (0.05)* | 0.60 (0.18) |
| Messenger Summary | 0.38 (0.03) | 0.33 (0.07) | 0.47 (0.03) | 0.30 (0.07) |
| Messenger Summary + Vision | 0.37 (0.04) | 0.38 (0.05) | 0.48 (0.04) | 0.33 (0.09) |
| Messenger Summary + VoT | 0.38 (0.05) | 0.37 (0.06) | 0.38 (0.04)* | 0.33 (0.03) |
| BabyAIGoTo Summary | 0.52 (0.51) | 0.96 (0.20) | 0.44 (0.51) | 0.60 (0.52) |
| BabyAIGoTo Summary + Vision | 0.36 (0.50) | 0.96 (0.20) | 0.72 (0.46) | 0.40 (0.52) |
| BabyAIGoTo Summary + VoT | 0.12 (0.33) | 0.60 (0.50)* | 0.56 (0.51) | 1.00 (0.00) |
| BabyAIOpen Summary | 0.00 (0.00) | 0.00 (0.00) | 0.00 (0.00) | 0.40 (0.52) |
| BabyAIOpen Summary + Vision | 0.08 (0.19) | 0.00 (0.00) | 0.00 (0.00) | 0.30 (0.48) |
| BabyAIOpen Summary + VoT | 0.04 (0.20) | 0.32 (0.48)* | 0.04 (0.20) | 1.00 (0.00)* |
| BabyAIPickUpSeqGoTo Summary | 0.08 (0.28) | 0.32 (0.48) | 0.16 (0.37) | 0.30 (0.48) |
| BabyAIPickUpSeqGoTo Summary + Vision | 0.08 (0.28) | 0.48 (0.51) | 0.24 (0.44) | 0.60 (0.52) |
| BabyAIPickUpSeqGoTo Summary + VoT | 0.00 (0.00) | 0.32 (0.48) | 0.20 (0.41) | 0.70 (0.48) |
| BabyAIPickUp Summary | 0.12 (0.33) | 0.24 (0.44) | 0.20 (0.41) | 0.10 (0.32) |
| BabyAIPickUp Summary + Vision | 0.12 (0.23) | 0.28 (0.46) | 0.12 (0.33) | 0.30 (0.48) |
| BabyAIPickUp Summary + VoT | 0.00 (0.00) | 0.68 (0.48) | 0.24 (0.44) | 1.00 (0.00)* |
| BabyAIPutNext Summary | 0.00 (0.00) | 0.00 (0.00) | 0.04 (0.20) | 0.00 (0.00) |
| BabyAIPutNext Summary + Vision | 0.00 (0.00) | 0.00 (0.00) | 0.00 (0.00) | 0.00 (0.00) |
| BabyAIPutNext Summary + VoT | 0.00 (0.00) | 0.00 (0.00) | 0.00 (0.00) | 0.10 (0.32) |

## 4.3 Spatial grounding

We finally examine whether augmenting the state with additional spatial information improves in-context decision-making. For these experiments, we use SmartPlay's default state interfaces: *TaggedList* for Tower of Hanoi and *NaturalLanguage* for Messenger (see Appendix for examples), and the summarised historical trajectory. We treat this text-only prompt as the baseline and compare it to two spatially grounded variants, the first is adding an image of the state, and the second is adding a text-based spatial map generated via VoT. Table 6 reports the normalized performance and Figure 3 visualises the performance differences relative to the text-only baseline.

### 4.3.1 Hanoi

Table 6 shows that adding raw pixel input *Vision* is often unhelpful and can be detrimental: LLaVA-PHI3-3.8B significantly degrades under *Summary+Vision* (0.16 to 0.11), and QWEN3-VL-32B-INSTRUCT also drops markedly (0.65 to 0.47). Effects are otherwise weak or inconsistent (QWEN2.5VL-7B: 0.33 to 0.33; LLaVA-7B: 0.19 to 0.25), suggesting that images are mostly unhelpful.

*VoT* is more beneficial. *Summary+VoT* improves mid-scale models (Qwen2.5VL-7B: 0.33 to 0.38; LLaVA-7B: 0.19 to 0.39*) and slightly improves LLaVA-Phi3-3.8B (0.16 to 0.18). For example, comparing LLaVA-7B runs with and without the VoT protocol under the same summary setting illustrates this effect. With VoT, the model maintains an explicit intermediate spatial state (e.g., *"Map (Top-Down View)"* and *"Map Update Notes"*), producing a coherent sequence of valid moves and reaching a higher final score (2). Without VoT, responses frequently exhibit action–rationale mismatches. For instance, the reasoning proposes moving a disk from A→B while the selected action corresponds to B→A, and the episode ends with score 0. This comparison suggests that explicitly externalising the state can improve alignment between the model's reasoning and the executed action.

However, *VoT* is not uniformly helpful (Qwen3-VL-32B-Instruct: 0.65 to 0.60), and several models fail to consistently construct a good map, as illustrated by the following example from LLaVA-Phi3-3.8B.

```
Map (Top-Down View):
| A | B | C |
---+---+---+---
A | 0 | 1 | 2
B | 3 | 4 | 5
C | 6 | 7 | 8
```

Qualitative analysis shows that *VoT* only helps when the model can reliably generate a map that preserves the task-relevant structure accurately, which is especially difficult for smaller models.

### 4.3.2 Messenger

In Messenger, spatial grounding yields no consistent gains across models. *Vision* produces small, mixed results relative to the baseline (LLaVA-Phi3-3.8B: 0.38 to 0.37; Qwen2.5VL-7B: 0.33 to 0.38; Qwen3-VL-32B-Instruct: 0.30 to 0.33), with only a marginal but significant improvement for LLaVA-7B (0.47 to 0.48). *VoT* is similarly model-dependent and can reduce performance: while it improves Qwen2.5VL-7B (0.33 to 0.37) and Qwen3-VL-32B-Instruct (0.30 to 0.33), it significantly degrades LLaVA-7B (0.47 to 0.38*) and leaves LLaVA-Phi3-3.8B unchanged (0.38 to 0.38).

Qualitative results show that *VoT* often produces unreliable maps that are fabricated or internally inconsistent (e.g., LLaVA-7B draws # grids while describing them as both "walls" and the "dangerous airplane," and uses labels such as E as both "east" and an object identifier). In addition, we observe frequent role/synonym errors (e.g., labeling the goal ship as "neutral" while treating other entities as enemies), and even when models state the correct rule ("need the message"), they may still navigate to the goal without it and terminate.

### 4.3.3 BabyAI

In BabyAI, the usefulness of spatial grounding depends strongly on task demands. In simpler navigation tasks such as GoTo, *VoT* significantly degrades Qwen2.5VL-7B, reducing performance from 0.96 to 0.60*. Yet the same representation can strongly help larger models, with Qwen3-VL-32B-Instruct improving from 0.60 to 1.00 under *VoT*.

In contrast, tasks with explicit interaction and manipulation demands show clearer gains from structured grounding. In Open, *Vision* provides little benefit and can degrade performance (e.g., Qwen3-VL-32B-Instruct drops from 0.40 to 0.30), while *VoT* yields large, significant improvements (e.g., Qwen3-VL-32B-Instruct improves from 0.40 to 1.00*; Qwen2.5VL-7B improves from 0.00 to 0.32*). The effect is even more pronounced in PickUp: *Vision* is modest (e.g., Qwen2.5VL-7B improves from 0.24 to 0.28), whereas *VoT* can unlock large gains, improving Qwen2.5VL-7B from 0.24 to 0.68 and Qwen3-VL-32B-Instruct from 0.10 to 1.00*.

However, this advantage does not carry to all BabyAI tasks. In PickUpSeqGoTo, *Vision* is often the more reliable augmentation (e.g., Qwen3-VL-32B-Instruct improves from 0.30 to 0.60), while *VoT* can be neutral or harmful for weaker models (e.g., LLaVA-Phi3-3.8B degrades from 0.08 to 0.00). Finally,

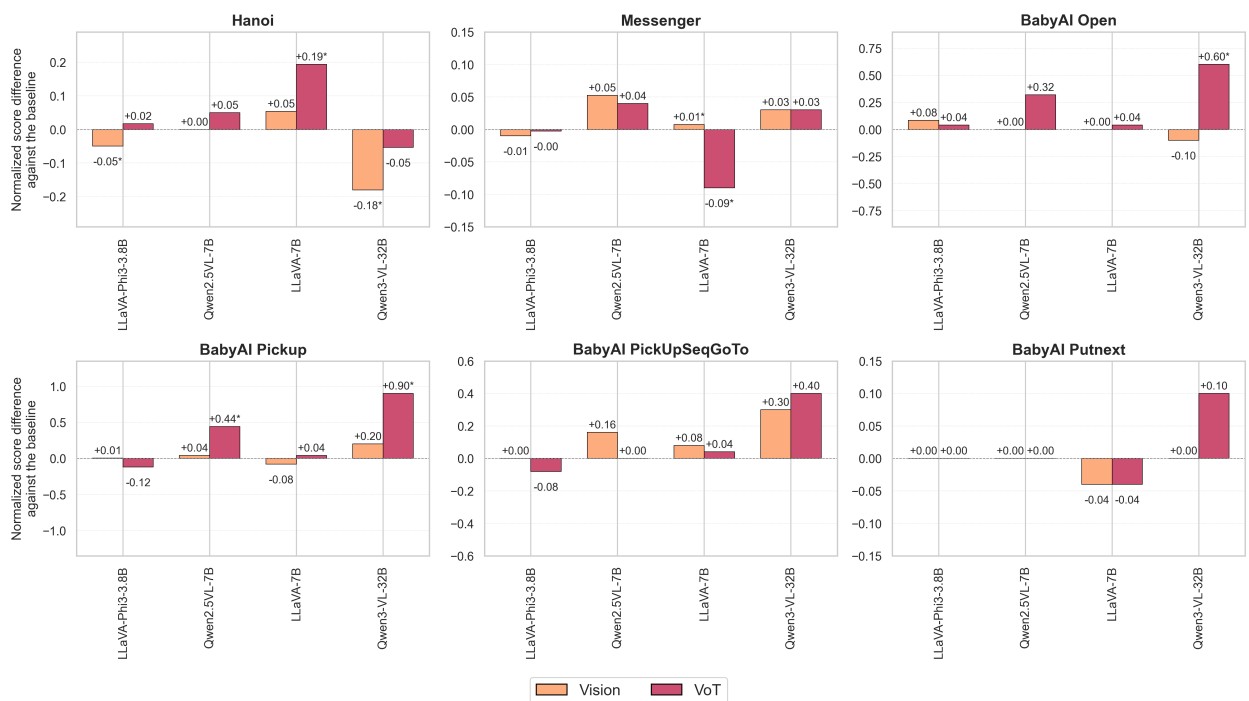

Figure 3: Effect of spatial grounding on performance relative to the text-only baseline. We report normalised performance differences from adding spatial grounding via images (*Vision*) or text-based spatial maps (*VoT*); positive values indicate improvement and negative values indicate degradation, while near-zero values indicate no measurable change (neutral). A superscript * marks a significant difference from the baseline for the same model and task, using a run-level bootstrap test of the mean difference over per-run scores (10,000 resamples; percentile 95% CI). Two-sided bootstrap p-values are adjusted with the Holm procedure across all pairwise comparisons, controlling the family-wise error rate at $\alpha = 0.05$. Overall, *VoT* improves over baseline in 15/24 instances, degrades in 5/24, and is neutral in 4/24; *Vision* improves in 10/24, degrades in 6/24, and is neutral in 8/24. Significant improvements are from *VoT* in Hanoi for LLAVA-7B, in BabyAI OPEN for QWEN3-VL-32B-INSTRUCT, and in BabyAI PICKUP for QWEN2.5VL-7B and QWEN3-VL-32B-INSTRUCT; *VoT* also shows a significant degradation in Messenger for LLAVA-7B. For *Vision*, significant effects are mostly degradations in Hanoi for LLAVA-PHI3-3.8B and QWEN3-VL-32B-INSTRUCT, with only a marginal significant improvement in Messenger for LLAVA-7B.

PUTNEXT remains near floor across representations, indicating that explicit spatial grounding alone does not resolve the longer-horizon relational reasoning required.

Larger models benefit most from *VoT*: for GOTO, OPEN, and PICKUP it lifts performance to a perfect score (e.g., QWEN3-VL-32B-INSTRUCT: 0.60 to 1.00; 0.40 to 1.00*; 0.10 to 1.00*), and in PICKUPSEQGOTO it approaches optimal performance (0.30 to 0.70), suggesting that constructing and exploiting a useful spatial map requires sufficient model capacity.

Qualitatively, failures in harder BabyAI tasks follow a small set of recurring patterns: agents attempt interactions from too far away, struggle to decompose instructions into stable substeps (find → pick up → navigate → place), lose track of held objects or completed subgoals, and fail to maintain precise relative positioning (e.g., "next to"). Overall, spatial grounding helps in BabyAI when it directly supports interaction-level control, but it does not overcome limitations in long-horizon relational planning.

Table 7: Oracle VoT ablation on BabyAIPickUp. We compare *Summary*, *Summary + VoT*, and *Summary + VoT Oracle* (programmatic ground-truth ASCII map), reporting mean normalised score $\pm$ SD over 25 runs. *+VoT* improves performance only for models with sufficient reasoning capacity, whereas providing ground-truth maps alone yields no consistent gains. This suggests that the benefit comes less from access to spatial information and more from the act of constructing a spatial representation as a reasoning scaffold. A superscript * marks conditions that differ significantly from *Baseline*, using a bootstrap test of the mean difference over per-run scores (10,000 resamples; 95% CI); to address multiple comparisons, p-values are adjusted with the Holm–Bonferroni method ($\alpha = 0.05$) across all pairwise comparisons, controlling the family-wise error rate.

| Environment | LLaVA-Phi3 | | | LLaVA-7B | | | Qwen2.5VL-7B | | |
|---|---|---|---|---|---|---|---|---|---|
| | Baseline | +VoT | VoT Oracle | Baseline | +VoT | VoT Oracle | Baseline | +VoT | VoT Oracle |
| BabyAI-PickUp | 0.12 (0.32) | 0.00 (0.00) | 0.08 (0.27) | 0.20 (0.40) | 0.24 (0.43) | 0.24 (0.43) | 0.24 (0.43) | 0.68 (0.47)* | 0.16 (0.37) |

## 4.4 Ablation: Visualization-of-Thought oracle

The *Visualization-of-Thought* (*VoT*) condition relies on the LLM to generate a structured spatial sketch from observations. This conflates two factors: whether an explicit spatial abstraction is useful for decision-making, and whether the model can reliably construct such an abstraction. To disentangle these effects, we introduce an *Oracle VoT* ablation in the BABYAI-PICKUP task.

We compare three conditions: (1) LLM-generated *VoT*, where the agent produces its own ASCII map; (2) oracle-generated ASCII maps, constructed programmatically from the true environment state; and (3) the text-only baseline. This comparison isolates whether failures under *VoT* stem from the usefulness of spatial grounding itself or from errors in map construction.

Across models, providing ground-truth spatial maps yields no consistent benefit over the text-only baseline, whereas LLM-generated VoT can substantially improve performance, but only for models with sufficient reasoning capacity.

In particular, QWEN2.5VL-7B improves markedly from 0.24 (Baseline) to 0.68* under LLM-generated VoT, yet drops to 0.16 when given oracle maps, performing worse than baseline despite access to perfect spatial information. LLAVA-7B shows no reliable differences across conditions ($0.20 \rightarrow 0.24 \rightarrow 0.24$), while LLAVA-PHI3 fails to benefit from either intervention, degrading from 0.12 (Baseline) to 0.00 under VoT and 0.08 under Oracle VoT.

These results suggest that performance gains do not come from spatial information alone. The benefit of VoT appears to arise from the process of constructing a spatial representation: generating an explicit map forces the model to reason sequentially about object locations. This effect depends strongly on model capability. Weaker models struggle to generate reliable spatial maps, producing outputs that can be uninformative or wrong. Stronger models use the act of generation to make spatial relationships explicit and improve control. However, perfect spatial maps do not improve performance. This indicates that the main challenge is not having access to the correct map, but performing the step-by-step reasoning to understand the state. Therefore, VoT functions as a mechanism to facilitate spatial reasoning rather than a simple description of the state.

In addition, providing an oracle map can lead to a representation–model mismatch. Although oracle maps contain correct spatial information, their specific ASCII layout and symbol conventions may be out-of-distribution for the model as an input format, making them harder to parse and exploit. In contrast, when the model constructs the map itself, it can choose token patterns and conventions that align with its learned priors, and condition on more reliably. Under this view, VoT helps not only by externalizing spatial relations, but by allowing the model to express those relations in a format closer to the structured-text patterns it has seen during training.

### 4.4.1 Cross-task comparison

As shown in Fig. 3, adding spatial grounding generally improves over the text-only baseline, with *VoT* yielding more frequent gains than adding images (*Vision*). However, *VoT* is only reliable when the model is sufficiently capable to construct and use a consistent spatial map; smaller models often produce incomplete or incorrect maps, limiting benefits or causing regressions.

Importantly, the gains from *VoT* are not simply due to exposing additional spatial information. Providing perfect spatial maps does not improve performance, indicating that the key benefit comes from the act of constructing the map itself. This suggests that *VoT* functions as a mechanism to elicit step-by-step spatial reasoning, rather than a static description of the state.

## 5   Conclusion and discussion

In this study, we examined how the inference-time representation of the state, expressed along the axes of granularity (long-form trajectories versus summaries), structure (natural language versus concise symbolic encodings), and spatial grounding (text-only representations versus explicit spatial grounding via images or textual map encodings) influences the reasoning capabilities of LLMs in dynamic environments.

Trajectory summarisation is the most broadly useful intervention. Replacing the full interaction history with a rolling summary often improves long-horizon decision-making by reducing redundancy and removing noise. This also highlights the difficulty current models face in processing long temporal sequences. Maintaining the interaction history presents a trade-off. Long-form trajectories preserve complete information but increase the number of tokens processed at each timestep, leading to higher inference costs and latency. Conversely, summarised trajectories reduce context length and computational overhead, but require an additional summarisation steps and risk omitting task-relevant information. The benefit depends on whether compression preserves task-critical details, and summarisation alone cannot lift performance when the underlying task demands exceed the model's capabilities.

Across tasks, *Natural Language* remains the most robust representation and is consistently competitive across model families. When structured encodings help, models with stronger exposure to code or structured outputs are generally more tolerant of code-like schemas (e.g., JSON-style formats) and can exploit them to reduce relational inference, whereas other models become brittle under the same representations and often fail to interpret the schema reliably.

Adding spatial information can further improve performance, and among the spatial grounding variants, *VoT* is the most consistently useful. However, these gains require sufficient model capability to reliably construct a faithful map; weaker models often generate noisy or incorrect sketches. Notably, providing perfect spatial maps does not improve performance over the text-only baseline, indicating that the main challenge is not access to the correct map but carrying out the step-by-step reasoning needed to use the state effectively. In this sense, *VoT* functions primarily as a mechanism that elicits spatial reasoning, rather than as a richer description of the state.

Furthermore, spatial grounding remains a bottleneck that is not automatically resolved by multimodal architectures. Contrary to the common intuition in embodied AI that direct visual access should enable better physical reasoning (Driess et al., 2023; Reed et al., 2022), we find that providing an image often reduces performance relative to text-only baselines. One possible explanation is that VLMs under-utilise image tokens during generation ("visual neglect") (Chen et al., 2025), making pixel inputs a noisy and weakly-informative state interface in long-horizon control.

Together, these results highlight that how the state is represented matters as much as what information it contains. State representations do not merely transmit information to the model, but actively shape the form of reasoning that the model performs. Compact representations are beneficial when they preserve task-critical structure while removing irrelevant information, but can be harmful when they omit details required for decision-making. Similarly, structured or spatial representations are only effective when the model is sufficiently capable of reliably interpreting or constructing them; otherwise, they introduce noise or mismatches that degrade performance.

Reflecting on the criteria for effective state representations described in 2.1, our findings suggest that the main challenge is to provide a state that is compact while still capturing the full context needed to choose actions and predict their long-term success. Summaries and structured encodings can reduce context length, but might omit details necessary to evaluate the quality of the current situation. Conversely, richer natural-language states better preserve semantic information and support generalisation to new scenarios, at the cost of longer context. As such, the encoding format is not a guarantee of quality; effectiveness must be proven by measuring the actual performance of the downstream controller (Lesort et al., 2018).

Future research should consider the following. First, we observed that failures often arise from drift in the LLM-maintained state representation of the summary trajectory or VoT, where small omissions, hallucinated facts, or internal inconsistencies accumulate over long horizons and eventually influence downstream decisions. A promising direction is therefore to pair the agents with verifiers during the process. This can be done through self-verification (Madaan et al., 2023; Shinn et al., 2023; Wang et al., 2022) or external verification using tools for (sub)tasks that involve math or formal rules (Chen et al., 2024b; Gao et al., 2023). Second, we observe that current VLMs remain weak at temporal spatial state tracking: even when object recognition is reliable, models often fail to maintain and update a coherent spatial belief over long sequences. Current pretraining regimes rely mostly on image–text pairs that rarely require precise spatial discrimination or persistent geometry across time (Chen et al., 2024a). Progress may therefore require training signals that directly encourage maintaining a consistent spatial state over many steps. For example, combining explicit geometric supervision with interactive trajectories that force the agent to repeatedly update its beliefs under partial observability (Chen et al., 2024a; Song et al., 2025). Third, in this study, we have compared two trajectory settings: either maintaining the full interaction history or using a fully summarised history, and we have not considered hybrid memory schemes. For example, summarising earlier steps while retaining a window of more recent context. Evaluating such hybrid trajectory designs is an important direction for future work. Fourth, our findings are based on a limited set of open-source models and benchmarks, and we have not evaluated how these representational trade-offs manifest in proprietary models. Extending the analysis to a broader range of task families and environments remains future work, because real-world environments are dynamic (Plaat et al., 2025), static or one-shot benchmarks can overestimate competence by avoiding a key challenge of interactive decision-making: maintaining and updating state under (partial) observability as new evidence arrives. We therefore argue for a shift toward dynamic evaluation, as true agentic competence is not defined by one-shot accuracy, but by the ability to dynamically reconcile new evidence with an internal world model to navigate the unpredictability of the physical world.

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

# A    Appendix

---

**Example 1: Agent Prompt**

You are an intelligent agent with the goal of succeeding in the game by maximizing the cumulative score. Your decisions should be based on the game manual, the current observation, and your past trajectory.
# **Task:**

1. Analyze the current observation and past trajectory to select the most suitable action that aligns with the game's objectives and maximizes cumulative rewards.

2. Choose one action from the provided list of actions, starting from index 1, and provide a concise reason for your choice.

# **Response Constraints:**

1. Select only one action from the provided list.

2. Provide reasoning that directly links the chosen action to the game's objectives and observed patterns.

3. Respond strictly with the action and the reason in the specified format.

# **Response Format:**
Action: [action number]. Reason: [explanation]
# **Input data**
Game Description: {manual}
Current observation: {obs}
Past trajectory: {Full trajectory} OR {Summarized trajectory}
Question: {question}

---

**Example 2: Summarization Prompt**

Your role is to compress the recent trajectory into a concise summary that an agent can reuse next step.
Game Description: {manual}
Recent history (most recent last): {recent_history}
Previous rolling summary: {previous_summary}
**Task:**

- Produce a new summary ≤25 tokens.

- Mention agent location, key items/inventory, goals/hazards, and momentum toward objectives.

- Do not restate every step; only the most salient facts.

- If the history is empty, return "Start of game".

- Do not only repeat an action, but summarize what actions led to what outcomes.

Respond strictly in this exact format. The summary should be one sentence only, but you can use a comma.
Summary: <concise summary here>

---

**Example 3: Visualization-of-Thought Prompt**

You are an intelligent agent whose objective is to maximize cumulative score in the game. Ground every decision in the game manual, the current observation, and your remembered trajectory.
# **Visualization-of-Thought Protocol**
You will:

- Draw a compact top-down ASCII map of the current situation before choosing an action.

- Update that map as you reason about candidate moves, annotating notable changes (e.g., planned path, hazards, inventory).

- Decide the single best next action from the provided list.

**Guidelines for the ASCII map:**

- The ASCII map must represent only what is explicitly visible in the observation.

- Keep it ≤6×6 unless the observation explicitly demands more detail.

- Use consistent symbols (e.g., `A` for agent, `G` for goal, `#` for wall, `.` for empty) and include a legend when helpful.

- If multiple hypothetical moves are considered, show intermediate marks such as `?` or arrow glyphs to illustrate your updates.

- If the environment is non-grid-based (e.g., Hanoi), draw the most compact symbolic state instead of a grid.

# **Required Output Sections**

1. **Map (Top-Down View):** Latest ASCII grid after incorporating reasoning updates.

2. **Map Update Notes:** Bullet list (≤3 bullets) summarizing how the map changed during reasoning.

3. **Reasoning:** Brief chain-of-thought tying observation/manual to the chosen action.

4. **Action:** `Action: [number] ([action name]). Reason: [concise justification].`

5. **Summary:** Finish with `Summary: <≤25-token recap of agent location, inventory, hazards, and next intent>` so the rolling summary stays current.

# **Live Input**
Game Description: {manual}
Current observation: {obs}
Past trajectory: {Full trajectory} OR {Summarized trajectory}
Question: {question}

---

**Example 4: Hanoi:NaturalLanguage**

Peg A has disk 2 at the bottom, disk 1 in the middle, and disk 0 on top. Peg B is empty. Peg C is empty.

**Example 5: Hanoi:DictList**

'A': [2, 1, 0], 'B': [], 'C': []

**Example 6: Hanoi:Matrix**

[[2, 1, 0], [-1, -1, -1], [-1, -1, -1]]

**Example 7: Hanoi:TaggedList**

- A: |bottom, [2, 1, 0], top|
- B: |bottom, [], top|
- C: |bottom, [], top|

**Example 8: Messenger:NaturalLanguage**

You took action Move North.
You (agent) already have the message.
You see:
- bird 9 steps away
- ship 9 steps away
- sword 5 steps away

**Example 9: Messenger:NaturalLanguagePos**

You are an agent with the message. You are currently in position 5, 6. You can see a mage 3 steps to the west, a dog 1 step to the east, a ball 3 steps to the northwest.

**Example 10: Messenger: Coordinates**

```
COORDINATE SYSTEM:
Agent:  (5, 5)
Entities:
airplane_0:  (5, 3)
ball_0:  (7, 5)
queen_0:  (3, 5)
```

**Original View:**
You (agent) don't have the message.

You see:
– airplane 2 steps to your west
– ball 2 steps to your south
– queen 2 steps to your north

**Example 11: Messenger:Symbolic**

```
..........
..........
..........
..........
..........
...G.A.M..
..........
.....E....
..........
..........
Legend:
A=agent(no msg)
P=agent(with msg)
.=empty
Entities:
E=fish
M=scientist
G=robot
```

