# OpenReview forum: "State Design Matters: How Representations Shape Dynamic Reasoning in Large Language Models"
_TMLR — Accepted by TMLR_

### Review · Reviewer_8XVq · 2026-02-28

**Summary Of Contributions:**

This paper studies how inference-time state representations shape LLM/VLM performance in dynamic, multi-step environments, holding model parameters fixed and varying only the prompt-level state interface. The authors organize "state design" along three axes: (i) granularity (full interaction history vs summarized trajectory), (ii) structure (natural language vs symbolic encodings such as dictionaries/matrices), and (iii) spatial grounding (text-only vs image inputs vs textual map encodings). Empirically, the paper evaluates open-source LLMs/VLMs on Hanoi & Messenger (SmartPlay) and BabyAI tasks (BALROG) using multiple representation variants, and reports normalized score improvements and bootstrap-based significance testing. In this paper, a representation refers to the inference-time encoding used to translate the environment state into the model's input context. Different representations contain similar information but structure it differently, thereby inducing different reasoning behaviors in the same underlying model. The paper presents a systematic study that shows that representation choice can substantially influence performance. In other words, changing the interface language to a fixed intelligent system changes its apparent reasoning ability.

Other more specific findings include:
* Trajectory summarization often improves long-horizon performance, especially when the task admits a compact abstraction (e.g., Hanoi), but can harm when critical decision details are omitted.
* Natural language appears the most robust representation across model families, while structured encodings can help mainly for models with strong code/structured-output priors, and can be brittle for others.
* Spatial grounding via images is generally inconsistent and often degrades performance, suggesting that images rarely provide useful additional grounding beyond the text state. VoT tends to be more consistently beneficial than alternative spatial grounding variants.


Strengths
* Clear research question and controlled/rigorous methodology: varying state representations while keeping model weights fixed and comparing across multiple tasks/models.
* Strong use of oracle-style ablations to separate interface utility from model ability.
* Inclusion of qualitative failure modes.
* Supplementary material provides concrete prompt templates (agent prompt, summarization prompt, VoT protocol) and examples of the representations used, supporting reproducibility.
* The authors provide code at https://anonymous.4open.science/r/state-representations-llms-dynamic-tasks-87BA/README.md

Weaknesses
* The authors perform many comparisons (tasks $\times$ models $\times$ representations), but I think there is no control of the Family-Wise Error Rate (FWER) and/or explicit discussion about multiple comparisons (potentially inflating false positives).
* The paper does not include an introductory explanatory figure that would help readers quickly understand the scientific framing of the work. In fact, the first figure only appears on page 11.

**Audience:**

Yes

**Audience Explanation:**

The findings are directly relevant to researchers working on LLM agents, prompt and interface design, sequential decision-making with foundation models, context compression and memory mechanisms, and multimodal reasoning evaluation. Importantly, the paper provides actionable insights about representation choices without proposing new architectures, aligning well with TMLR's emphasis on technically sound empirical understanding rather than novelty.

**Broader Impact Concerns:**

From my point of view, the work does not present direct ethical risks.

**Claims And Evidence:**

Yes

**Claims Explanation:**

The main claims of the paper are supported by empirical evidence obtained through controlled experimental comparisons in which model parameters are held fixed while only the state representation interface is varied. This design directly targets the central research question and reduces confounding factors related to model scale or training differences.

Across multiple environments (SmartPlay and BabyAI) and model families, the authors report performance differences associated with representation choices, supported by repeated runs and bootstrap-based confidence interval estimation. Importantly, the paper complements performance results with oracle-style ablations (e.g., Oracle Summary and Oracle VoT), which help disentangle improvements arising from information availability versus representation construction.

The claims made in the manuscript are generally descriptive and empirically grounded, rather than speculative or beyond the presented evidence. While some interpretative conclusions could benefit from additional clarification, the experimental results presented are sufficient to support the core empirical findings reported in the paper.

**Requested Changes:**

Below, I propose several changes or potential improvements to the paper, none of which are critical. That is, in my view, the paper as it currently stands is reasonably robust.

1) The paper reports statistical significance using bootstrap confidence intervals computed independently for each comparison. However, given the large number of model $\times$ task $\times$ representation comparisons, no explicit control of the family-wise error rate or false discovery rate is provided. Clarifying that would strengthen statistical interpretability.

2) I believe the paper would benefit from a figure in the Introduction to visually illustrate its main methodological proposal and/or scientific goals. In particular, I refer to a figure that would allow readers, at a glance, to understand what the authors aim to do at the methodological level.

3) Regarding the experimental setup and the choice of LLMs, if possible, it would be beneficial to provide some justification or rationale explaining why those specific LLMs were selected, as well as the reasoning behind the chosen hyperparameter values (e.g., "temperature 0.2 and nucleus sampling with top_p=0.95"). Are the results sensitive to these values?

4) I think it would be positive to include a clarification regarding the statistical methodology used for bootstrap confidence interval estimation. The paper reports bootstrap-based confidence intervals to assess statistical significance of performance differences across representations. However, it is not entirely clear what constitutes the resampling unit in the bootstrap procedure. In particular, are confidence intervals computed by resampling runs or individual episodes within runs? In my opinion, this distinction is important because, I assume, episodes executed within the same run share the same model configuration, prompting setup, and experimental conditions, and may therefore exhibit non-negligible correlations. If resampling is performed at the episode level rather than at the run level, the independence assumption underlying bootstrap estimation may be violated. Correlated samples effectively reduce the number of independent observations, causing bootstrap variance estimates to be underestimated and confidence intervals to appear overly narrow (i.e., optimistic). Could the authors clarify the level at which bootstrap resampling is applied, and whether independence between resampled units is assumed or empirically justified?

5) I believe it would be helpful to introduce certain concepts before mentioning them as if any reader were already perfectly familiar with them. For instance, VoT (Visualization-of-Thought). On the other hand, it is not entirely clear how performance is measured. What performance metrics are computed? Where exactly do the reported scores come from? While the paper frames evaluation in terms of episodic cumulative reward in an MDP, the manuscript does not specify how this reward (or any benchmark-specific success signal) is converted into the reported normalised score in each environment. Providing an explicit mathematical definition of the evaluation metric would substantially improve reproducibility and interpretability.

6) If I'm not mistaken, representations are enforced only through prompt construction. In that case, would it be possible that performance differences arise from prompt-induced behavioural changes unrelated to representation fidelity? Have the authors considered verification analyses (e.g., trajectory ablations or information masking) to confirm reliance on the provided representation?

7) The results suggest relevant model-representation interactions. Why do certain models interact better with specific representations? For instance, the paper suggests that models trained on code handle JSON formats (DictList) better, but a more detailed discussion of these pre-training strategies would add significant value.

---

> ### Author Response · Authors · 2026-03-21
> **Point-by-point response to Reviewer 8XVq**
>
> Thank you for your valuable feedback. We address each requested change below.
>
> 1. Thank you. We now apply the Holm procedure to all two-sided bootstrap p-values within each table, controlling the family-wise error rate at alpha = 0.05. We revised the tables, results, and captions accordingly.
> 2. We have added Figure 1 to the Introduction. The figure visually summarises the main idea of the paper, illustrating how different state representations are constructed.
> 3. Thank you for this comment. We have added clarifications to Section 3.4 regarding the choice of LLMs and hyperparameters: *We evaluate a set of open-source LLMs to study how model scale and architecture affect performance. The models were selected to cover a range of parameter scales and architectural families, including both text-only and vision–language models. This allows us to examine whether the observed representation effects are consistent across heterogeneous models rather than being specific to a single architecture.*  Regarding hyperparameters: *All experiments use temperature 0.2 with nucleus sampling top_p = 0.95 to reduce stochastic variability while avoiding fully deterministic decoding.* Since our tasks require consistent state tracking, a low temperature minimises unnecessary variability, while nucleus sampling prevents strictly greedy decoding.
> 4. We now clarify that bootstrap resampling is performed at the run level, not the episode level, in Section 3.5: *For all comparisons against the baseline, we assess significance using a run-level bootstrap test of the mean difference. For each run, we first average the benchmark-specific episode scores to obtain a single run-level score. We then generate 10,000 bootstrap replicates by resampling these per-run scores with replacement and compute percentile 95% confidence intervals for the mean difference, and significance markers are based on Holm-corrected two-sided bootstrap p-values*
> 5.  We revised the manuscript to define Visualization-of-Thought at first mention, and have added the following to the Introduction: *"Visualization-of-Thought" (VoT), a prompting method that elicits spatial reasoning by having models visualize intermediate reasoning traces, suggests that pixel inputs are not always necessary: structured text layouts such as ASCII grids can be sufficient for spatial tasks (Wu et al., 2024a).* We now explain in Section 3.5 how scores are computed:
>
> Performance is evaluated using the task-specific episode score $s_e$ at the end of evaluation episode $e$. In Tower of Hanoi, $s_e$ is the number of disks placed on the goal peg, with values in $[0,3]$. In Messenger, $s_e$ is the episode reward, with values in $[-1,1]$. In BabyAI, $s_e$ is binary, where $1$ indicates successful task completion and $0$ otherwise. For each run, we average scores over $E$ evaluation episodes:
> $$
> s_{\mathrm{run}} = \frac{1}{E} \sum_{e=1}^{E} s_e
> $$
> where $E$ is the number of evaluation episodes per run.
> We then report the mean over $R$ runs:
> $$
> \bar{s} = \frac{1}{R} \sum_{r=1}^{R} s_{\mathrm{run},r}
> $$
> where $r$ indexes runs and $s_{\mathrm{run},r}$ denotes the mean score of run $r$.
> To enable comparison across environments, we linearly rescale the mean score $\bar{s}$ to a normalized score $z \in [0,1]$, where $0$ denotes the worst outcome and $1$ denotes successful task completion. For Tower of Hanoi, we use
> $$
> z = \frac{\bar{s}}{3},
> $$
> and for Messenger, we use
> $$
> z = \frac{\bar{s} + 1}{2}.
> $$
> BabyAI scores already lie in $[0,1]$, so no additional normalization is required. Standard deviations are transformed using the same linear scaling.
>
> 6. Thank you for this comment. The task manual, action space, and observation are identical across conditions; only the state representation varies through prompt construction. This isolates the effect of representation itself. The consistency of these effects across environments and model families also makes differences driven by prompt wording a less likely explanation
>
> 7. We expanded Section 4.2.3 to connect our findings with the models' public training details. *These interactions appear correlated with a model’s ability to treat the prompt as a structured data interface. Models with stronger evidence of code or structured-output training (e.g., DeepSeek-R1-14B and Qwen2.5VL-7B) are generally more robust to JSON-like schemas, while others degrade under the same format, suggesting schema interpretation failures rather than task difficulty. Technical reports for the Qwen2.5 family emphasize coding, structured data extraction, JSON-style outputs, and tool use, which may make Qwen-based models better able to interpret prompts such as DictList. Consistent with this view, DeepSeek-R1-14B, distilled from Qwen2.5, may inherit similar structured-output priors. By contrast, documentation for Llama 3.1, and LLaVA emphasizes dialogue, reasoning, or multimodal instruction following rather than explicit schema-oriented outputs.*

---

> > ### Comment · Reviewer_8XVq · 2026-03-22
> >
> > Dear all,
> >
> > I have read the other reviewers' comments and the authors' response to my comments, and I must admit that, at least with regard to my questions and proposed modifications, the authors' response is satisfactory.
> >
> > At this point, I would not have any further comments, constructive criticisms, or questions for the authors.
> >
> > Best.

---

### Review · Reviewer_2SXt · 2026-03-04

**Summary Of Contributions:**

The paper studies how the state representation in prompts has impact on the performance of LLMs on multi-step reasoning. The paper systematically studies three aspects of state representation, which are history granularity (full trajectory or summary), structure (natural language or symbolic encodings), and spatial grounding (text-only or images or text-based maps). The paper conducts experiments on environments including Tower of Hanoi, Messenger, and BabyAI. The results show that representation choices significantly change performance of LLMs when the model is unchanged. In summary, the experiment results indicates that full trajectories, natural language representations and text-based spatial maps can improve long reasoning, encouraging step-by-step spatial reasoning instead of simply providing more information.

**Audience:**

Yes

**Audience Explanation:**

The paper is relevant to researchers working on LLM agents, prompting methods, and sequential decision-making, so the results on improved prompt representations could help make the usage of LLM-based systems more effective.

**Broader Impact Concerns:**

There are no impact concerns.

**Claims And Evidence:**

Yes

**Claims Explanation:**

The claims are generally supported by empirical evidence. The experiments directly compare the three proposed representation methods against baselines. It would be beneficial to further evaluate these methods on additional non–grid-based tasks and larger models (e.g., GPT-5.2 or Claude-4.6).

**Requested Changes:**

1. Evaluate the proposed representation methods on a broader set of tasks beyond grid-based environments.
2. Include experiments with additional large frontier models to test whether the observed trends hold at larger model scales.
3. Add a short analysis or qualitative examples to better explain why certain representations work better for some models.
4. Since LLMs have memory limits, discuss the trade-offs between performance gains and inference cost.

---

> ### Author Response · Authors · 2026-03-21
> **Point-by-point response to Reviewer 2SXt**
>
> Thank you. We address each requested change below.
>
> 1.  Thank you for this suggestion. Our evaluation already goes beyond purely grid-based tasks: Tower of Hanoi is a planning task without a grid, and Messenger also tests semantic comprehension and synonym understanding. We agree that broader evaluation would be valuable, but running the full experimental suite across additional environments and multiple LLMs was not feasible within the revision timeframe. We have clarified this limitation in Section 5, Conclusion and Discussion, and note broader task families and proprietary models as future work:
>
> *Fourth, our findings are based on a limited set of open-source models and benchmarks, and we have not evaluated how these representational trade-offs manifest in proprietary models. Extending the analysis to a broader range of task families and environments remains future work.*
>
> 2.   We agree that evaluating larger frontier models would be valuable. However, extending the full experimental suite to proprietary, API-based models would require substantial additional computation and cost, which was not feasible within the rebuttal window. We therefore acknowledge this limitation in Section 5, Conclusion and Discussion, as noted in point 1, and leave evaluation on broader model classes as future work.
>
> 3. We added qualitative analyses illustrating failure modes and reasoning patterns under different state encodings, to clarify why some representations work better for specific models. We added qualitative examples to Section 4.1.2:
>
> *The summarisation failures largely depend on what information is retained. Models often produce generic summaries. For example, Llama3.1-8B writes “Agent has not obtained the message yet, and there are entities nearby (scientist, ship, robot)”, while omitting critical details such as relative distances, outcomes of recent actions, or progress toward subgoals. Failures also arise from reasoning limitations, such as choosing to “do nothing” even when the message holder is one step away. Qualitative traces further indicate that the benefit of summarisation is model-specific and depends on whether the model can maintain a coherent internal state. For instance, with DeepSeek-R1-14B on Messenger, adding summaries stabilises the trajectory: without summaries the agent oscillates (“Move North → Move West → Move East → Move West”) and finishes with score 0, whereas with summaries (e.g., “Agent moves north toward the dog to get the message”) the same model reaches score 1 in a short rollout.*
>
> Section 4.2.2:
>
>  *In the Messenger environment, models that rely on explicit reasoning traces often perform best when observations are expressed in NaturalLanguage. For example, given the description “You see a bird two steps west, a fish two steps north, and a sword two steps south”, DeepSeek-R1-14B correctly identifies the sword as the message and reasons that moving south will reach it. The textual description directly exposes entity names and relative spatial relations, allowing the model to reason over the environment in a way that resembles natural language instructions.*
>
> And Section 4.3.1:
>
> *For example, comparing LLaVA-7B runs with and without the VoT protocol under the same summary setting illustrates this effect. With VoT, the model maintains an explicit intermediate spatial state (e.g., “Map (Top-Down View)” and “Map Update Notes”), producing a coherent sequence of valid moves and reaching a higher final score (2). Without VoT, responses frequently exhibit action–rationale mismatches. For instance, the reasoning proposes moving a disk from A→B while the selected action corresponds to B→A, and the episode ends with score 0. This comparison suggests that explicitly externalising the state can improve alignment between the model’s reasoning and the executed action.*
>
> 4.    Thank you for this comment. Representation choices affect prompt length and, therefore, the computational cost of inference. Long-form trajectories preserve all historical information but increase the number of tokens processed, which leads to higher latency and cost. Summarised trajectories reduce token usage and improve efficiency at the prompt level, but may omit task-relevant information and thereby degrade performance. Moreover, maintaining a summary requires an additional LLM call to compress the trajectory, introducing extra computational overhead. We have clarified this trade-off in the manuscript and added the following to Section 5, *Conclusion and Discussion*:
>
> *Maintaining the interaction history presents a trade-off. Long-form trajectories preserve complete information but increase the number of tokens processed at each timestep, leading to higher inference costs and latency. Conversely, summarised trajectories reduce context length and computational overhead, but require additional summarisation steps and risk omitting task-relevant information.*

---

### Review · Reviewer_Q1qS · 2026-03-09

**Summary Of Contributions:**

This paper systemically studies the design principles of state representations in LLMs, from three aspects: state granularity, state structure, and spatial grounding. The authors draw conclusions from experiments in multi-step, interactive environments. The conclusions indicate the importance of history summarization, the robustness of natural language representation, and the effectiveness of Visualization of Thought.

**Audience:**

Yes

**Audience Explanation:**

The paper systemically studies the designs of state representations, which can inspire future LLM agents methods and frameworks.

**Claims And Evidence:**

Yes

**Claims Explanation:**

The authors draw conclusions from multiple multi-step, interactive environments where the state evolves in response to the agent’s actions and success requires long-horizon planning, memory, and spatial reasoning.

**Requested Changes:**

1. Realistic agentic LLM setups can be different (and more complicated) from the setups considered in this paper. For example, it is very popular to use the combination of the trajectory summarization and the long-form trajectories. In other words, the agents may compact the earlier histories and append all of the recent context. Such differences may weaken the conclusion of the paper.
2. For the same state representation, different state designs may also impact the conclusions. For example, the effectiveness of summarisation strongly depends on the quality of the generated summary. If important details are omitted or distorted, performance can degrade, making the approach unreliable in some tasks.

---

> ### Author Response · Authors · 2026-03-21
> **Point-by-point response to Reviewer Q1qS**
>
> Thank you for your comments. Below, we address each of your comments point by point.
>
> 1. **Comment:**
>    *Realistic agentic LLM setups can be different (and more complicated) from the setups considered in this paper. For example, it is very popular to use the combination of trajectory summarization and long-form trajectories. In other words, the agents may compact the earlier histories and append all of the recent context. Such differences may weaken the conclusion of the paper.*
>
>    **Response:**
>    We thank the reviewer for this helpful point. In this paper, we compare only two settings, *Long Form* and *Summary*, to isolate the effect of state granularity under controlled conditions. This makes the representational comparison easier to interpret, but it does not cover the broader space of hybrid setups. We have therefore revised the manuscript to clarify that our conclusions are limited to the controlled settings studied here, and we discuss hybrid trajectory designs as an important direction for future work in Section 5, *Conclusion and Discussion*:
>
>    *Third, in this study we have compared two trajectory settings: either maintaining the full interaction history or using a fully summarised history, and we have not considered hybrid memory schemes. For example, summarising earlier steps while retaining a window of more recent context. Evaluating such hybrid trajectory designs is an important direction for future work.*
>
> 2. **Comment:**
>    *For the same state representation, different state designs may also impact the conclusions. For example, the effectiveness of summarisation strongly depends on the quality of the generated summary. If important details are omitted or distorted, performance can degrade, making the approach unreliable in some tasks.*
>
>    **Response:**
>    We thank the reviewer for this important observation. We agree that the effectiveness of summarisation depends not only on the compressed state representation, but also on the quality of the generated summary. The model must both generate a rolling summary of the trajectory and use that summary for decision making. This conflates two factors: whether a compressed state is a useful interface for control, and whether the model can reliably produce a faithful summary of the environment state.
>
>    To disentangle these effects, we include the *Oracle Summary* ablation. In this setting, the agent is provided with the ground-truth summary, removing errors introduced by LLM-generated summaries. The results show that for some models, such as Llama3.1-8B and DeepSeek-R1-14B, performance improves further under *Oracle Summary*, indicating that imperfect summaries can introduce information loss. In contrast, weaker models such as LLaVA-Phi3 do not benefit even when the summary is perfect, suggesting that both summary quality and the model’s ability to act upon that information are important.

---

### Decision · Action_Editor_pzh6 · 2026-04-08

**Recommendation:** Accept as is

**Audience:**

Yes

**Audience Explanation:**

Yes. The paper is of interest to a reasonable number of people in TMLR's audience.

**Claims And Evidence:**

Yes

**Claims Explanation:**

Yes. The claims made in the submission are supported by accurate, convincing and clear evidence.